# Does the Internet Bring People Closer Together or Further Apart? The Impact of Internet Usage on Interpersonal Communications

**DOI:** 10.3390/bs12110425

**Published:** 2022-10-31

**Authors:** Chao Li, Guangjie Ning, Yuxin Xia, Kaiyi Guo, Qianqian Liu

**Affiliations:** 1Business School, Shandong University, Weihai 264209, China; 2HSBC Business School, Peking University, Shenzhen 518055, China

**Keywords:** Internet, interpersonal communications, family interactions, contacts with friends, loneliness

## Abstract

The complementarity interference (CI) model suggests that the Internet may either inhibit or facilitate interpersonal communications. This paper empirically examines the impact of Internet usage on interpersonal interactions, using a micro dataset from China to answer whether the Internet brings people closer together or further apart. The empirical results demonstrate, first, that Internet usage significantly increases both the time and frequency of people’s communications with their family and friends, rather than causing them to feel more disconnected and isolated. Holding other factors constant, for each one-standard-deviation increase in Internet usage, weekly communications with family members increases by an average of 102.150 min, while there is an average increase of 54.838 min in interactions with friends. These findings as to its positive effects are robust when using other regression models and interpersonal contact measures, as well as the instrumental variable method. Second, Internet usage also contributes to decreased loneliness; it exerts this effect primarily by improving people’s interactions with their family members. However, communications with friends do not significantly mediate such impacts. Third, the positive role of Internet usage on communications is more prominent for people with more frequent online socialization and self-presentation, better online skills, younger age, higher educational level, and who are living in urban areas. In addition, the beneficial effects of Internet usage are larger for communications with family members in the case of migrants. Therefore, in the context of the rapid development of information technology, the network infrastructure should be improved to make better use of the Internet to facilitate interpersonal communications and promote people’s wellness.

## 1. Introduction

Whether Internet usage brings people closer together or further apart is an important but unanswered question. With the rapid development of information technology, the Internet has been widely used in various areas almost all over the world. According to Internet World Stats, compared with the year 2000, the number of global Internet users in 2022 has increased by 14.16 times. By 31 July 2022, there were 5.47 billion Internet users in the world out of the 7.93 billion global population, and the penetration rate has steadily risen to 68.98% [1]. The rapid taking up of the Internet has profoundly changed human society in multiple aspects. On the macro level, it has reduced transaction costs, promoted industrial upgrading [2,3], and driven economic development [4,5]. On the micro level, the Internet has tremendous impacts on people’s daily lives and has changed their lifestyles, habits, attitudes and preferences [6,7,8]. However, the impact of Internet usage on interpersonal communications is still unclear. The complementarity interference (CI) model suggests that the Internet may either inhibit or facilitate interpersonal interactions.

On the one hand, in terms of the interference aspects, there may be a crowding-out effect of Internet usage on interpersonal contacts. Using the Internet may divert people’s attention from communicating with family and friends to other activities, such as playing online games, surfing websites, watching online videos, and live streaming. This may lead people to ignore real-life communications [9,10,11]. In addition, excessive addiction to the Internet can also trigger people’s depression, anxiety, and emotional impulsivity, resulting in a poor psychological state and even social phobias [12,13]. This may also cause people to reduce their interpersonal interactions. Based on this finding, it has been hypothesized that the more time spent on the Internet, the fewer interpersonal communications there will be.

On the other hand, in terms of complementarity, the Internet may facilitate interpersonal communications by reducing communication costs and providing opportunities for teleconferencing. Before the popularization of modern information technologies, people could only communicate by meeting face to face. Later, the development of communication technologies, such as the telegraph and telephone, eliminated the geographical boundaries of interpersonal contacts and made remote communication a reality. However, traditional communication technologies can only transmit information via voice and text messages and have the drawback of high cost. The Internet has greatly reduced the cost of communications, shortened the distances between disparate groups, and has even made it possible for people to meet via video conferencing [14]. In addition, Internet technologies have brought a variety of emerging communication platforms, such as Facebook, WhatsApp, Zoom, and WeChat, helping people to communicate more conveniently at a much lower cost [15,16,17]. On the basis of this evolution, it is hypothesized that the more time people spend on the Internet, the more interpersonal communications there are.

Interpersonal communications are essential to building social networks, which is also a necessary channel to help people establish social trust and enhance their sense of belonging and happiness [18,19,20]. Therefore, in the context of the rapid development of Internet technology, it is of great importance to clarify the impact of the Internet on interpersonal communications. If Internet usage can facilitate interpersonal interactions at a lower cost and in a more convenient way, then we should make full use of this technology to promote communications. Conversely, if the Internet reduces interpersonal communications, then necessary measures should be taken to alleviate its negative effects on interpersonal interactions while utilizing the benefits of the Internet in other aspects. Therefore, this paper aims to empirically examine the impact of Internet usage on interpersonal communications, using the Chinese General Social Survey. The robustness and endogeneity of the results are also tested from multiple perspectives. On this basis, we further explore the impact of Internet usage on people’s feelings of loneliness and the mediating role of interpersonal communications. In addition, the heterogeneities of the Internet’s effects are systematically investigated.

Compared with the existing literature, the contributions of this paper are mainly reflected in two aspects. First, this paper enriches the research concerning the Internet’s impacts on people. Most of the existing literature examines the influence of the Internet from the points of view of working conditions, psychological states, emotions, health, preferences, and lifestyles [20,21,22,23,24], while little research has been conducted concerning its effect on interpersonal communications. Second, this paper deepens our understanding of the influencing factors of interpersonal interactions, from the new perspective of modern information technology. Existing studies in the field of social communications have mainly focused on the effects of demographic characteristics, social identities, culture, and so on [25,26,27], lacking any assessment of the impact of the Internet.

This paper aims to examine the impact of Internet usage on interpersonal communications, as well as to investigate the heterogeneities in its effect, to systematically answer whether the Internet brings people closer together or pushes them further apart. This study is carried out following the research logic of “literature review—theoretical framework—empirical tests—further discussion—heterogeneity analysis”. A systematic literature review is given in Section 2. Based on the literature, a theoretical framework using the complementarity interference (CI) model is presented in Section 3, wherein the hypotheses are proposed. To test the theory, the data, variables, and empirical results are presented in Section 4 and Section 5. Section 6 further discusses the impact of Internet usage on loneliness and the mediating role of interpersonal communications, as well as the Internet’s effects in other respects. Section 7 explores the heterogeneities of the impact of Internet usage. Section 8 summarizes all the conclusions drawn in the above sections, Section 9 identifies the theoretical and practical implications, and Section 10 discusses the study’s limitations and further research directions.

## 2. Literature Review

### 2.1. The Impact of Internet Usage on People’s Lives

With the advancement of information technology, the Internet has become more and more indispensable in people’s daily lives. The Internet has brought tremendous positive impacts in multiple aspects. For example, Internet-based telecommuting is becoming a convenient and increasingly popular mode of work around the world [28]. Moreover, studies have found that self-presentation on social media helps users to achieve higher psychological well-being [29]. Heterogeneity analysis demonstrates that the positive impact of self-presentation on social media on psychological well-being is more significant in those with higher self-esteem [30]. Through online comparisons, people are able to generate benign envy, which is helpful for inspiration [22,31].

However, it has also been found that the Internet has mixed and heterogeneous impacts on its users. For example, while for girls, daily Internet use was not associated with higher levels of depressive symptoms [32], for boys, a positive association between the two factors did exist [33]. Interestingly, a study based on a Chinese sample found a significantly positive association between Internet use and mental health [34]. Many studies have also identified jealousy as one of the main symptoms of poorer states of mental health resulting from Internet use [35,36,37]. In addition to psychological health, existing studies find a significant negative association between mobile Internet use and self-rated health [34]. Moreover, online games are one of the most significant applications of the Internet and their impacts are controversial. It is found that frequent exposure to violent online games tends to be associated with an increase in aggressive behavior, desensitization, and physiological arousal, while also showing a decrease in empathy [38]. However, other studies have found that the correlation between online games and aggressive behaviors is not significant [23,24]. In addition, practical games are widely used in multiple areas of education, healthcare, sustainability projects, training, and consultancy, but their effectiveness varies due to differences in the designs [39,40,41,42,43].

Furthermore, Internet usage has also led to the emergence of Internet addiction, a new clinical disorder [44]. The COVID-19 pandemic has further increased people’s Internet online usage and a rising prevalence of Internet addiction has been reported among people in various occupations [45,46]. Although Internet addiction has not been recognized by the World Health Organization (WHO) and the American Psychological Association (APA), existing studies have shown that it is a new type of serious mental disorder [47]. There are heterogeneities in the severity and prevalence of Internet addiction. Regionally, Internet addiction has a greater impact on Internet users in developed areas, such as in Europe and the United States [48,49]. Studies also found that those with greater neuroticism are more likely to become addicted to the Internet [50,51]. Heterogeneity also exists in terms of gender, age, and social class. For example, people with a higher social class are less likely to experience Internet addiction [49,52].

### 2.2. Factors Affecting Interpersonal Communications

Interpersonal communication is a complex social process and is closely related to people’s well-being. Evidence shows that those with a high level of communication skills have a better mental health status compared to their counterparts [53,54]. Other studies have found that higher interpersonal stress is associated with stronger symptoms of insomnia, which, in turn, is associated with poorer mental health status [55,56]. In addition to its important impact on the psychological well-being of individuals, interpersonal communication also plays an essential role in building strong family relationships [57,58]. The importance of interpersonal communication is also reflected in many other aspects, including improving learning ability, obtaining job opportunities, promoting career development, etc. [59,60].

Regarding the factors affecting interpersonal communications, studies have found that age, gender, culture, social background, working characteristics, geographical distance, and technology exert a level of influence [25,58,61], although there is disagreement about the exact impact of these factors. For example, ethnic background affects interpersonal communications to some extent, mainly because people with different backgrounds are more likely to experience cultural misunderstandings with each other [26]. It has already been mentioned above that interpersonal communication can influence mental health and physical activities. Likewise, the two factors also affect interpersonal interaction. A study using a sample of college students found that social anxiety had a negative impact on their interpersonal communication skills, while psychological resilience played a mediating role between them, and perceived social support from teachers and classmates further moderated their psychological resilience [62]. It was also found that physical activity can facilitate family communication among family members because it provides more opportunities for them to meet [27,63].

In general, existing studies demonstrate that interpersonal communications are of great importance in promoting people’s mental health and helping families to build resilience. At the same time, interpersonal communications are conducive to acquiring new knowledge and playing a better role in both the family and society. Moreover, demographic, work, human capital, and social characteristics are the main factors that influence interpersonal communication.

### 2.3. Possible Relationship between Internet Usage and Interpersonal Communication

As mentioned above, interpersonal communication plays an important role in people’s lives, work, and careers; nowadays, it can be achieved by face-to-face interactions as well as via the Internet. At the same time, the Internet has both pros and cons in many aspects. So how does the Internet impact interpersonal communications? Based on existing research, it is believed that frequent exposure to the Internet distracts users from their offline lives [64]. For example, the use of mobile Internet via smartphones distracts parents from spending time with their children and undermines the communication between parents and children [65]. Furthermore, another study shows that children’s Internet use is also associated with a decrease in their participation in family activities. When people are overly dependent on the Internet, online activities can replace offline social connections with their family members and friends [11,66]. Internet addiction has also been proven to lead to a reduction in people’s social and interpersonal skills [47,67], which may further reduce their communications with family and friends. Although the effect of Internet usage on interpersonal communications has not been directly studied, the aforementioned studies imply that time spent on the Internet may crowd out interpersonal interaction, to some extent.

However, other studies point to the possible positive effects of Internet usage on interpersonal communication. Thanks to the development of Internet technology, today, text messages and voice calls are no longer the main methods for people of all ages [15,16]. Social networking software and group chats have become popular communication platforms [27,68]. Many studies have found that the use of the Internet effectively brings much convenience to interpersonal connections for both the young and old cohorts [69,70], which in turn can benefit people’s well-being [71]. Indeed, compared with traditional communication methods, such as letters, telegraphs and phone calls, the Internet provides innovative means of communication, such as video meetings, in a more convenient and cost-saving way. For example, WhatsApp has been shown to facilitate intergenerational family interactions [17]. Facebook helps to maintain interpersonal relationships for those who have difficulty making social connections, especially for people with low self-esteem [72].

Overall, the Internet has changed people’s lives tremendously, although its effects on interpersonal communications have not been systematically tested. In this context, it can be hypothesized from the existing research that the Internet may crowd out interpersonal communications [47,64,65,66,67]. Nevertheless, many studies believe that the Internet reduces the cost of communication between people, offering more diverse and convenient ways to make contact [15,16,68,69,70,71,72]. Therefore, it can also be speculated that the Internet may shorten the distances between people, thereby promoting interpersonal communication. However, even with theoretical analysis and the existing literature, the impact of Internet usage on interpersonal interactions is still unclear. In view of this gap in the literature, we aim to systematically investigate how the Internet affects interpersonal communication.

## 3. Theoretical Framework

### 3.1. Internet Usage

This paper aims to investigate the impact of Internet usage on interpersonal communication. For the explanatory variable, time spent on the Internet is the most direct and important indicator by which to measure Internet usage; it is very intuitive and is widely used [11,67,73]. In addition, it has been applied not only to characterize how people generally use the Internet in their daily life but also to measure possible excessive Internet use and Internet addiction [74].

### 3.2. Interpersonal Communications

Studies have shown that communications with family members and friends are most important in people’s daily interpersonal interactions [75,76,77]. In the benchmarking analysis, time spent on communications with family and friends is used to reflect interpersonal interactions. Meanwhile, considering that the frequency of interactions is also a very important indicator for interpersonal contact, this is used for further robustness analysis. Both kinds of indicators have been applied to measure the levels of interpersonal contact in existing research [78,79,80,81].

The complementarity interference (CI) model [53,82] of the Internet, as illustrated in Figure 1, provides a theoretical framework for analyzing the relationship between Internet usage and interpersonal communication. Based on the following theoretical analysis, Internet usage may either facilitate or deteriorate interpersonal communication.

### 3.3. Interference Aspects of Internet Usage

#### 3.3.1. Distracting Attention

Studies have confirmed a significantly negative correlation between online and offline activities [83]. The Internet may divert people’s attention away from interpersonal interactions to online activities, including video games, online news, short videos, live streaming, etc., leading users to neglect communications with family and friends in real life [10,66]. This suggests that Internet usage may shift people’s attention; there may be a crowding-out effect of Internet usage on interpersonal communication. In addition, research has also found a negative correlation between Internet usage and time spent accompanying family members [11]. Although they do not specifically examine how the Internet affects family communications, the findings imply that time spent online may reduce interpersonal interactions inside the family, to some extent. In addition, in parent–child contact, the parent’s attention is easily distracted by online activities via smartphones, resulting in compromised parent-child bonds [65].

#### 3.3.2. Reducing Social Skills

It has been shown that people tend to establish fewer offline social networks when they are overly dependent on the Internet [68]. This may be due to the fact that Internet usage reduces people’s social and interpersonal skills [67], thereby decreasing their communications and interactions. Studies have also found that among adolescents with a higher prevalence of Internet addiction, social skills are generally poorer [47]. Moreover, Internet addiction is proven to be closely related to attention deficit disorders, further causing social phobia [84]. Another study has identified that inadequate social skills and social fears decrease interpersonal communication [85]. Consequently, Internet use may hinder interpersonal interactions by reducing people’s social skills.

#### 3.3.3. Increasing Negative Emotions

Internet usage may trigger depression, anxiety, and impulsiveness in some people, resulting in poor psychological states and negative emotions [10]. This may also further lead to a decrease in interpersonal interactions [11,73]. In addition, compared with face-to-face communications, Internet-based interpersonal interactions are disadvantaged in terms of emotional transmission and are, thus, less effective in enhancing effective communication [53,83]. Another source of negative feelings brought about by the Internet is peer pressure. Nowadays, people tend to share their daily lives via online platforms, inadvertently causing them to make comparisons with the lives of others. This makes people more pessimistic about their body image and standard of living, resulting in increased anxiety [86]. The nervousness caused by peer pressure on the Internet leads users to be more reluctant to communicate with others in the real world.

Based on the interference aspects of Internet usage, Hypothesis 1 can be proposed:

**Hypothesis** **1.**The more hours people use the Internet, the less time they spend on interpersonal communications.

### 3.4. Complementarity Aspects of Internet Usage

#### 3.4.1. Reducing the Cost of Interpersonal Communications

Before the popularization of modern information technologies, people could only communicate face-to-face. Later, the telegraph and telephone eliminated the geographical boundaries of interpersonal contact and made remote communication a reality [87]. However, traditional communication techniques face the problem of high costs. The Internet has greatly reduced both the time and money needed for instant communication, narrowed the distances between people, and made simultaneous communication affordable. For example, compared with telephone calls, Internet-based voice calls and online meetings cost much less in time and money for people to communicate [14,88].

#### 3.4.2. Enriching Communication Channels and Modes

Traditional communication technologies mainly transmit voice and text, but it is difficult for them to simulate face-to-face interactions. The Internet has spawned a variety of emerging communication channels and modes, such as Facebook, WhatsApp, Zoom, and WeChat, which can help people to replicate face-to-face interactions more realistically online [15,16]. For example, during the COVID-19 epidemic, various network platforms facilitated remote working and learning [89]. Without the Internet, this would have been almost unachievable. In addition, Internet-based communications help to improve the quality of people’s long-distance interactions compared to traditional methods. For example, it has been established that WhatsApp, an instant online messaging tool, can promote intergenerational communication among family members and help them build better bridges of understanding with each other [17].

#### 3.4.3. Building Wider Social Networks

The Internet helps people overcome communication barriers in real life, especially the fear of communicating with strangers, thereby establishing broader social networks [90]. Studies have found that Internet-based social networking platforms are effective in helping people share updates and, thus, build wider social connections across age, race, gender, geography, and social class boundaries [70]. Moreover, these enlarged social networks also create positive spillover effects in other aspects, improving people’s welfare. For example, people can use social media to communicate with others on health topics, which helps them become more health-conscious and intrinsically motivated to participate in physical exercises [69]. Therefore, online social connections contribute to improving people’s well-being, as well as promoting interpersonal communication and interactions [71,91].

Based on the complementarity aspects of Internet usage, Hypothesis 2 can be proposed:

**Hypothesis** **2.**The more hours people use the Internet, the more time they spend on interpersonal communications.

## 4. Data and Measures

### 4.1. Data Source

The data used in this paper come from the Chinese General Social Survey (CGSS), one of the most important and nationally representative academic surveys in China. The CGSS aims to systematically and comprehensively investigate the social and economic situations of the Chinese people. CGSS is part of the world General Social Survey group and the sampling of CGSS is based on a multi-stage stratified design. The National Survey Research Center at the Renmin University of China (NSRC) has organized the Chinese Social Survey Network (CSSN), including 49 universities and provincial social science academies. Detailed information regarding CGSS can be accessed via http://cgss.ruc.edu.cn/English/Home.htm (accessed on 25 September 2022). The reason for using CGSS is mainly due to its three advantages. First, CGSS surveys people’s interpersonal communication and the factors influencing it in the extension module, which is a convenient way to construct the explained variables and control variables. Second, CGSS contains information on the respondents’ habits of Internet usage, which facilitates the construction of an explanatory variable for this research. Third, CGSS contains the ISCO-2008 (International Standard Classification of Occupations, 2008) codes of the respondents’ occupations, which helps us construct an instrumental variable, based on job characteristics, to deal with the endogeneity problem. Since the key explained and explanatory variables used in this paper are only available in the extension module of CGSS in 2017, the 2017-wave dataset is used for this research.

### 4.2. Measures

The main explained variable in this paper is the time spent on interpersonal communication by the respondents. Communication with family members and friends is most important in people’s daily interactions [75,76]; therefore, we constructed indicators for communications with family and friends, denoted as “family communication” and “friends communication”, respectively. The two variables come from the following questions in CGSS’s extension module, “How many hours do you spend on communicating with your family per week on average?” and “How many hours do you spend on communicating with your friends per week on average?”, respectively. In the robustness analysis, other indicators of interpersonal communications were also constructed. The explanatory variable of this paper is the time spent using the Internet, denoted as “Internet usage”. This variable is derived from the respondents’ answers to the question: “How many hours do you use the Internet per week on average?”.

Based on the relevant literature concerning the factors influencing interpersonal communications [61,62], in order to avoid the bias of omitted variables, this paper controls those factors related to interpersonal communications as comprehensively as possible in the following six aspects. (1) Basic demographic characteristics, including gender, age and the squared term of age. (2) Working characteristics, including personal income, whether the participant is working in the system and whether they have a pension and medical insurance. (3) Human capital characteristics, including educational level and health status. (4) Social characteristics, including whether the participant belongs to any ethnic minorities, have certain religious beliefs, or if they are a Communist Party of China (CPC) member. (5) Family characteristics include family size and the number of children. (6) Regional characteristics include provincial dummies. Detailed descriptions and statistics of the above variables are given in Table 1.

## 5. Results

### 5.1. Benchmark Results

To investigate the impact of Internet usage on interpersonal communications, this paper first constructs the following ordinary least squares (OLS) benchmark econometric model.
(1)$$Interpersonal\_communication_{i}=\theta_{0}+\theta_{1}Internet\_usage_{i}+x_{i}^{\prime }\psi +d_{p}+\varepsilon_{i}$$

In model (1), $Interpersonal\_communication_{i}$ and $Internet\_usage_{i}$ represent the time spent on interpersonal communications and Internet usage, respectively, by the respondent, i. The time spent on communicating with family and friends is used to characterize $Interpersonal\_communication_{i}$. $x_{i}^{\prime }$ is the vector of the series of control variables described above. $d_{p}$ is the provincial fixed effect. This paper estimates the relationship between interpersonal communications and Internet usage with this model.

Table 2 shows the regression results, based on the above OLS model. Columns (1)–(3) demonstrate the results of the estimations concerning communicating with family members and columns (4)–(6) are estimated results concerning communicating with friends. It is clear that Internet usage is significantly and positively related to the time spent on communications with both family members and friends. Here, we conduct regression analysis by sequentially including the controls of different characteristics, with the aim of exploring whether the relationship between Internet usage and interpersonal communications is affected by other factors. Table 1 shows that, by gradually adding control variables from different aspects, the estimated coefficients of Internet use are stable at around 0.095 and 0.051 for the two explained variables, respectively. Moreover, all the estimates are significantly positive at the 1% level. This suggests that the more time people spend on the Internet, the more time they spend interacting with family and friends, supporting Hypothesis 2. It also means that the significant correlation between Internet usage and interpersonal interactions is not affected by other factors and is very robust. The above results prove that Internet usage does not lead to greater alienation among people. On the contrary, the Internet significantly enhances interpersonal communications.

In addition, the benchmark estimates also show that the effect of the Internet on interpersonal interactions is very notable. Holding other factors constant, for each one-standard-deviation increase in Internet usage (17.921 h per week), the weekly communication with family members increases by an average of 102.150 min (17.921 × 0.095 × 60), while there is an average increase of 54.838 min in interactions with friends. This demonstrates that while Internet usage has significantly positive effects on communications with both family and friends, it plays a more prominent role in facilitating interactions among family members.

### 5.2. Robustness and Endogeneity Checks

In order to examine the robustness of the relationship between Internet usage and interpersonal communications, and to tackle potential endogeneity problems, this paper conducts a series of robustness and endogeneity checks.

#### 5.2.1. Using the Poisson Model

Considering the fact that the dependent variables, which represent the time spent on communicating with family and friends, are discrete non-negative integers and fit the Poisson distribution, we use the Poisson model to conduct the robustness test. Table 3 shows that when using the Poisson model for communications with both family and friends, the estimated coefficients of Internet usage are all significantly positive at the 1% level. In addition, with the controlling characteristics as different aspects, the estimated coefficients of Internet usage fluctuate slightly but are generally very stable. This further confirms that our findings regarding Internet usage promoting people’s interpersonal communications do not rely on the selection of the OLS model.

#### 5.2.2. Using Other Indicators of Interpersonal Communication

In benchmark regression, we use the time spent on communications with family and friends to characterize interpersonal interaction. However, there may be measurement errors in some people’s perceptions of time. Furthermore, communication time may not adequately characterize the frequency of interpersonal communications. Based on this theory, to test the robustness of the findings, this paper further uses the frequencies of communication with family and friends as dependent variables, denoted as “Family communication frequency” and “Friends communication frequency”. These are derived from the respondents’ responses to “How often do you keep in touch with your family, on average?” and “How often do you keep in touch with your friends, on average?”. Answers are classified based on an eight-level scale from 1 to 8, representing “never”, “rarely”, “several times a year”, “once a month”, “2–3 times a month”, “once a week”, “several times a week”, and “every day”, respectively. Since they are ordered and explained variables for which the disparities between different levels of the scale are not equivalent, ordered Probit (Oprobit) and Logit (Ologit) models, as well as the OLS model, are used for estimation. The regression results are shown in Table 4. It is clear that when using these kinds of dependent variables to measure interpersonal communications, and no matter which model is applied, Internet usage has a significantly positive effect on the frequency of people’s interactions with family and friends, which further confirms the robustness of the findings.

#### 5.2.3. Endogeneity Tests

There may be endogeneity problems in the benchmark estimates, therefore, the significant relationship between Internet usage and interpersonal communications may be a correlation rather than causality. The endogeneity problems may result from two aspects, comprising reverse causality and omitted variable bias. Regarding reverse causality, we suggest that people may use the Internet more frequently because they are more willing to communicate with family members and friends. For example, individuals who live alone, who frequently travel and migrate, may use the Internet because of the need to communicate remotely with their friends and family. With respect to omitted variable bias, although we have controlled as comprehensively as possible those elements that affect interpersonal communications, there may still be factors that are difficult to characterize. In order to examine the causal relationship between Internet usage and interpersonal interactions and to tackle potential endogeneity problems, the following instrumental variable models are applied for carrying out further checks.
(2)$$Internet\_usage_{i}=\alpha_{0}+\alpha_{1}AI_{i}+x_{i}^{\prime }\psi^{1}+d_{p}+\varepsilon_{i}^{1}$$
(3)$$Interpersonal\_communication_{i}=\beta_{0}+\beta_{1}\hat{Internet\_usage_{i}}+x_{i}^{\prime }\psi^{2}+d_{p}+\varepsilon_{i}^{2}$$

$AI_{i}$ is the instrumental variable, which is the degree of artificial intelligence’s application in an individual, i’s, work. Model (2) performs first-stage regression, using $AI_{i}$ to estimate $Internet\_usage_{i}$. In model (3), second-stage regression is conducted to examine the effect of Internet usage on interpersonal communications, using the predicted values in the first-stage estimation. The $AI_{i}$ indicator comes from Mihaylov and Tijden [92]. Existing studies have shown that the higher the application of artificial intelligence in their work, the higher the requirements for people’s skills in using the Internet [93], and thus, the more likely they are to show increased Internet usage. Therefore, the instrumental variable satisfies the correlation requirement. In addition, since artificial intelligence is an exogenous technological change and is, thus, not related to micro individual characteristics, this variable satisfies the exogeneity condition. As shown in Table 5, results of the instrumental variable method with the two-stage least square (2SLS) method robustly prove that Internet usage has significantly positive impacts on interactions with family members and friends. This means that the significant relationship between Internet usage and interpersonal communications is causal rather than being a simple correlation.

#### 5.2.4. Missing Data Imputation

There are missing data in this research, with a missing rate of (3740−3507)/3740 = 6.223%. Although it seems that the missing rate is not high, missing data may cause sample selection problems, leading to biased and inconsistent statistical results, because the information may be missing but not at random. Considering that the dataset is cross-sectional rather than longitudinal and when referring to Ibrahim and Molenberghs [94], Kropko et al. [95], and Baraldi and Enders [96], we further tested whether the findings of this paper could be affected by the missing data problem, applying the following widely accepted approach. Specifically, we replace the missing values with the mean of the remaining values. Results using this approach are shown in Table 6 and it is clear that they are consistent with the benchmark estimations in this paper.

## 6. Further Discussions

### 6.1. Effects of Internet Usage on Loneliness

It has been confirmed in the sections above that Internet usage facilitates communications with family and friends. Furthermore, studies have shown that interpersonal communications are beneficial to increasing social support and reducing people’s loneliness [97,98,99]. Therefore, we are interested in whether Internet usage helps to reduce loneliness by increasing people’s interpersonal contacts. To test this hypothesis, we use an indicator to characterize loneliness, denoted as “Lonely”. It is taken from respondents’ answers to the question “I feel lonely”, which is based on the Likert scale from 1–5, representing “never”, “seldom”, “sometimes”, “often”, and “frequently”. The larger the values of the two variables, the higher the level of loneliness.

The first columns in Table 7 demonstrate the effect of Internet usage on loneliness, wherein the estimated coefficients of Internet usage are all significantly negative. This indicates that Internet usage significantly reduces loneliness. Meanwhile, columns (2) and (4) in Table 7 are the regression results of the impacts of Internet usage on communication with family members and friends, which are consistent with those in Table 3. Columns (3) and (5) present the results for when the indicators of family communication and friends communication are further included in regressions. The estimated results in column (3) of Table 7 show that communication with friends does not significantly affect people’s loneliness. However, in column (5), the estimates of family communication are significantly negative at the 1% level, implying that interactions with family help to decrease loneliness. At the same time, after the mediating variables, interpersonal communications are included in the regression, where the estimated coefficients of Internet usage remain significantly negative. Additionally, in column (5) of Table 7, the absolute values of the Internet usage estimates decrease, further proving that communication with family members plays a mediating role between using the Internet and loneliness. This implies that Internet usage reduces the feeling of loneliness by facilitating communication among family members. Family members are particularly important for Chinese people and the Chinese culture; therefore, relationships among family members have a more prominent impact on personal feelings [100]. Thus, compared with communication with friends, contacts with family members mediate the impact of the Internet in reducing loneliness more significantly.

### 6.2. Effects of Internet Usage in the Other Aspects

The above analysis shows the positive impact of the Internet on interpersonal communications, but it is not correct to assume that this usage has only a positive dimension. Further analysis using CGSS data, as shown in column (1) of Table 8, demonstrates that the more time people spend online, the easier it is to get addicted to the Internet, resulting in spending a longer time online than was planned. In addition, people who frequently use the Internet are more likely to feel anxious if they do not go online for a while (column (2) in Table 8). This is consistent with the existing studies, reporting that people tend to have difficulty controlling their time, and it is easier for them to become addicted to the Internet and the online world [45,46,47,48]. Furthermore, we find that going online reduces the amount of time people spend outdoors (column (3) in Table 8) and leads to more family complaints that they spend too much time online (column (4) in Table 8). Although we cannot directly verify the effect of Internet usage on face-to-face interpersonal communication, due to data availability, this is an indirect way to test whether Internet use reduces people’s face-to-face contact with the outside world and results in increased complaints from family members. Moreover, in terms of physical health, it was also found that more Internet usage also causes people to have worse eyesight (column (5) in Table 8), as well as neck and shoulder pain (column (6) in Table 8). The above analysis is based on six Likert 5-point scale variables from the responses to the question, “How do the following descriptions fit your situation?”: “I often spend more time online than I planned”, “If I don’t go online for a while, I will be anxious and restless”, “I spend less time outdoors because of using the Internet”, “My family complains that I spend too much time online”, “My eyesight has become worse because of using the Internet”, “I have neck and shoulder pain because of using the Internet”. Their responses are: “1—very untrue of me”, “2—untrue of me”, “3—neutral”, “4—true of me”, and “5—very true of me”. 

## 7. Heterogeneity Analysis

This paper further examines the heterogeneities of the impact of Internet usage on communications in different subgroups. First, in terms of the purposes of Internet usage, it is naturally hypothesized that if people use the Internet mainly for working or entertainment, rather than for interpersonal contact, then Internet usage should have no significant effect on their communications with family members and friends. This hypothesis is tested as follows. Specifically, this research divides the sample into subgroups, with different degrees of online social interactions and different preferences for online self-presentation, based on whether respondents frequently use social networking sites (including email, QQ, WeChat, Skype, etc.) to communicate with others, and whether they often post their updates on the social platforms (including WeChat, Moments, Qzone, Weibo, etc.). The regression results of Table 9 show that the impacts of Internet usage on communication with family and friends are only significant among those who often use the Internet to socialize, confirming the above hypothesis. In addition, posting updates regarding life and work via Internet social platforms also brings more online contacts. Table 10 shows that for individuals with a greater online presence, the positive effect of Internet usage on interpersonal communications is more pronounced. This means that for people who are more socially connected to the Internet, online activities significantly promote their interpersonal contacts. The heterogeneity results in this aspect also demonstrate that online social contact facilitates communications with family and friends and further confirm the robustness of the findings of this paper.

Furthermore, considering that communications via the Internet require certain online skills, it is naturally hypothesized that for individuals with better Internet skills, Internet usage should be more conducive to improving their interpersonal communication. This paper conducts a heterogeneity test for this hypothesis. According to whether the respondents are able to communicate with others proficiently online (the corresponding question in the CGSS questionnaire is: “Do you know how to express your thoughts and proficiently communicate with others online?”), the following subsample analysis is performed. The estimated results in columns (1) and (2) of Table 11 show that in terms of communications with family members, the positive effects of Internet usage are greater and are only statistically significant for those with more online skills. Columns (3) and (4) of Table 11 demonstrate that in terms of communications with friends, the role of Internet usage is significant for the two subgroups, but the estimated coefficient is larger for individuals skilled in online communications. This confirms that the impact of Internet usage on interpersonal contacts is more pronounced for people with better online skills.

Moreover, it has been shown in the existing literature that there are disparities in Internet usage and interpersonal communication among individuals of different ages and educational backgrounds [34,47,61]. Therefore, we further examine the heterogeneities of the impacts of Internet usage in the different subgroups, with different demographic characteristics. Table 12 shows that the impacts of Internet usage on communications with family and friends are significantly positive for both younger and older respondents. However, their effect is greater on the younger group under the age of 35, which may be due to the fact that young people are more inclined to use new online applications and are more skilled in Internet use. Therefore, the positive effect of Internet usage is more prominent in the younger cohort. The mean time of Internet usage for young individuals under 35 in CGSS is 23.59, which is much greater than that of their older counterparts, which is 7.56.

The results of the heterogeneity analysis in terms of education level are shown in Table 13. It is demonstrated that regardless of whether the respondents have a bachelor’s degree or above, the positive effect of Internet usage on interpersonal communications is significant. However, the Internet’s impact is more pronounced for those with higher educational levels. This may be due to the fact that the more educated groups have greater opportunities to learn and master the skills of using the Internet. In the CGSS sample, the average time of Internet usage among people with higher educational levels is much higher than the lower educated respondents (25.03 > 9.70).

In addition, in terms of regional heterogeneity, it is clear from Table 14 that the impact of Internet usage on communications with family and friends is more prominent for urban residents. Compared with their rural counterparts, urban residents are more familiar with the Internet in their work and daily life, due to faster technological development and better network infrastructure. Therefore, the descriptive statistics for the two subsamples show that the mean hours of Internet usage for residents in rural and urban areas are 16.29 and 8.64, respectively.

In addition, the Internet can help people to break geographical restrictions and realize remote communication, consequently shortening the distances between each other [19]. Therefore, it is natural to hypothesize that the role of Internet usage in facilitating communication may be more prominent for migrants. The regression results of Table 15 show that Internet usage has significant effects on promoting interpersonal communications, for both migrants and non-migrants. In particular, columns (1) and (2) show that in terms of family communication, the impact of Internet usage on migrants is more prominent than on non-migrants. However, columns (3) and (4) do not show a similar pattern in terms of communicating with friends. This is logical, since blood relationships among family members do not change due to migration, while friends can be found wherever you live. Migration leads to people moving further away from their families, geographically; consequently, the role of Internet usage in enhancing communications with family members is more prominent for migrants.

## 8. Conclusions

This paper empirically examines the impact of Internet usage on interpersonal communications with data from the Chinese General Social Survey to answer whether the Internet brings people closer together or further apart. The empirical results demonstrate that first, Internet usage helps to significantly increase the time and frequency of communications with family and friends, rather than causing people to feel more disconnected and isolated. This positive effect is robust when using various regression models and interpersonal contact measures, as well as the instrumental variable method. Specifically, the positive effects of Internet usage in promoting people’s interpersonal communications do not rely on the selection of regression models and are robustly significant regarding both the time that people spend on interactions, as well as the frequency of daily contacts. Furthermore, the relationship between Internet usage and interpersonal communications is proven to be causal rather than being a simple correlation, using the instrumental variable approach.

Second, Internet usage contributes to decreased loneliness, and it exerts this effect primarily by improving people’s interactions with their family members. However, communications with friends do not significantly mediate such impacts. This implies that the Internet reduces the feeling of loneliness by facilitating communication among family members, who are much more important in the Chinese culture, and therefore relationships among family members have a more important impact on personal feelings.

Third, the positive role of Internet usage on communications is more prominent for people with more frequent online socialization and self-presentation, higher online skills, younger age, higher educational levels and living in urban areas. In addition, the beneficial effects of Internet usage are larger on communications with family members for migrants. The reason may be that the blood relationships among family members do not change due to migration, while friends can be found anywhere.

## 9. Theoretical and Practical Implications

### 9.1. Theoretical Implications

This paper clarifies the net effect of Internet usage on interpersonal communications. Research has shown that Internet technology has tremendously enriched communication channels and modes [14,15,16,17,87,89]. Moreover, compared with traditional communication methods, such as phone calls and text messages, the Internet helps people to establish a much wider social network and achieve effective remote communication at a lower cost, as well as with greater efficiency [69,70,90,101]. Nevertheless, other studies reveal that Internet usage may distract people’s attention [10,11,65,66,83], reduce their social skills [47,67,68,84,85], and may even increase negative emotions [10,11,53,73,86]. The impact of Internet usage in this aspect would hinder interpersonal communications. No direct evidence is provided on how the Internet influences interpersonal communications. Therefore, according to theoretical analyses based on the existing literature, the net effect of Internet usage on interpersonal contacts is still unclear because of the coexistence of the complementarity and interference aspects. This research contributes to the literature by clarifying that the net effect of Internet usage on interpersonal communications is positive. The more that people use the Internet, the more they can interact with their family and friends. This positive effect is confirmed via various endogeneity and robustness checks. This paper shows that although the Internet may have both pros and cons, its overall impact is positive regarding interpersonal communication.

In addition, this paper further verifies the role of the Internet in reducing people’s loneliness, which is an important factor affecting well-being. Loneliness not only leads to depression but also reduces people’s life satisfaction and overall well-being [102,103]. Interpersonal interaction is an important element impacting loneliness [104]. Since Internet usage promotes communication, a natural question arises regarding whether it helps to decrease loneliness through this mechanism. If this speculation holds true, the robustness of the conclusions in this paper would be confirmed further. The existing research demonstrates that the Internet has enriched interpersonal communication channels [105]. Moreover, other studies reveal that interactions can help reduce loneliness, improve people’s well-being, and decrease depression [89,91,106]. In this paper, we present our findings that Internet usage lowers loneliness by promoting people’s communications with family and friends. Therefore, this study also contributes to the literature by elucidating the mechanisms underlying the well-being and emotional benefits of Internet use [107].

Furthermore, compared with previous studies supporting the positive effects of Internet usage [14,15,16,17,87,89], we also find heterogeneities in its impact from multiple perspectives. It is clear that not everyone gains equally from Internet use. The positive role of the Internet on interpersonal communication is more prominent for people with more frequent online socialization and wider self-presentation, better online skills, a younger age, higher educational levels, and who are living in urban areas. Some subgroups benefit more from Internet usage, while those who have been left behind in the digital age gain less. Heterogeneity analysis enriches the literature on the impact of the Internet, helping us to better identify vulnerable groups in the Internet era and create effective public policies accordingly.

### 9.2. Practical Implications

With the rapid progress of online technology, traditional face-to-face communication is gradually shifting toward social networking via the Internet as people are becoming immersed in the digital age. The Internet not only drives economic development but also helps people to interact with each other at a lower cost and in a more convenient way. The policy implications of this paper include the following recommendations.

First, the network infrastructure should be improved and updated to make better use of the Internet, to facilitate interpersonal communication among people. In the fast-changing world of information, the Internet has provided people with more and more convenient communication channels. We should continue to make better use of more advanced Internet technologies and improve the quality of the network, in order to enhance people’s online experience. Emerging technologies, such as 5G, should be applied to help people obtain more convenient and cheaper access to the Internet to improve their interpersonal communication and enhance social welfare.

Second, this paper reports that the Internet promotes interpersonal contact, thereby weakening people’s sense of loneliness. Therefore, establishing high-quality online communities via social networks is needed to help people enhance their well-being through further interactions. For those who suffer from loneliness, providing them with better access to the Internet may be an effective way to enhance their welfare. From the perspective of mental health, loneliness is related to an increased risk of mental disorders, such as depression, anxiety, and even dementia. Therefore, it is worth recommending that sufferers use the Internet to enhance their communications with others. For people with communication difficulties, online interactions can help them overcome their fear and help them to get in touch with others, thus establishing better social networks [90].

Third, policymakers should pay more attention to vulnerable subgroups in the Internet age, including older people and those with poorer online skills, those who are less well-educated, and those living in rural areas. These groups gain fewer benefits from Internet usage. Therefore, it is important to help them master the necessary online skills and provide them with more convenient and less expensive access to the Internet. For example, the network coverage should be extended to more remote and rural areas and the Internet connectivity there needs to be improved so that as many people as possible have equal access to the Internet. In addition, with the rapid development of Internet technology, individuals with lower education levels and older age may not be able to update their Internet skills. This may mean that they are unable to gain the benefits of Internet usage in terms of interpersonal communication. Therefore, in the context of the rapid application of emerging online technologies, enhancing the Internet skills of these vulnerable subgroups should be emphasized.

## 10. Limitations

First, since CGSS data is based on subjective answers, both the explanatory and explained variables in this paper are subjective indicators and there may, thus, be measurement errors caused by subjectivity. Although different variables are used as dependent variables in the robustness checks, confirming the positive effect of Internet usage on interpersonal interactions, these measures are also subjective. Therefore, we look forward to further testing the relationship between Internet usage and interpersonal contacts based on objective indicators in the future.

Second, as CGSS does not provide detailed information concerning the amount of time that people spend on the Internet for various purposes, we are unable to examine the effects of different types of online activities on interpersonal communications. In this regard, if people use the Internet mainly for working or for entertainment, rather than for interpersonal contacts, then online activities may well have a different effect on their communications with family members and friends. In the heterogeneity analysis, this research divided the sample into subgroups with different degrees of online social interactions and different preferences for online self-presentation. The results show that the impacts of Internet usage on communication with family and friends are only significant among those who habitually use the Internet to socialize and post updates. This indirectly examines the impact of different types of Internet usage on communications. We look forward to further investigating this issue in the future, on the basis of more detailed online data.

Third, this paper examines the impact of Internet usage on interpersonal communications in general. However, it is still not clear how Internet usage affects people’s face-to-face interactions. Due to data limitations, we are unable to directly test the quality of offline personal relationships, for example, changes in conversational topics, the willingness to broach topics discussed on the Internet, and the inclination to reveal true thoughts in a face-to-face relationship. The effects of Internet use on the quality of offline communications will be a very valuable research direction in the future.

## Figures and Tables

**Figure 1 behavsci-12-00425-f001:**
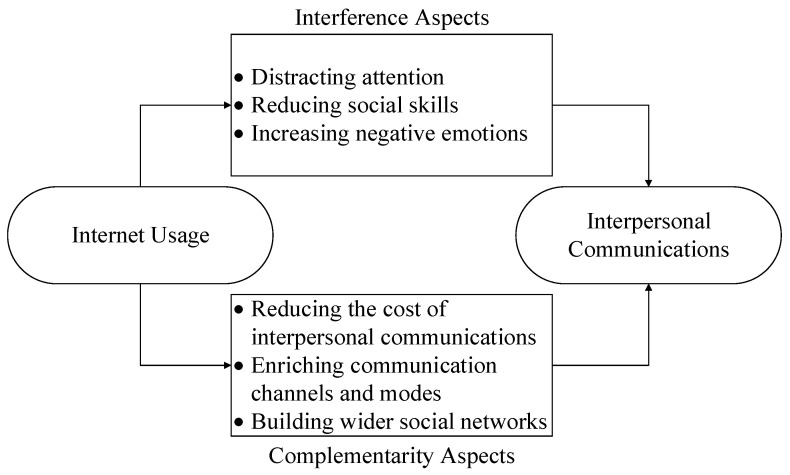
The theoretical framework of the study.

**Table 1 behavsci-12-00425-t001:** Summary statistics.

Variable	Description	Obs.	Mean	Std. Dev.	Min.	Max.
Dependent Variable						
Family communication	Number of hours communicating with family per week	3740	22.394	23.475	0	98
Friends communication	Number of hours communicating with friends per week	3722	7.467	9.874	0	98
Explanatory Variables						
Internet usage	Number of hours using the Internet per week	3857	11.405	17.921	0	98
Control Variables						
Demographic Characteristics						
Whether female	Yes = 1, No = 0	3857	0.515	0.500	0	1
Age	Age	3857	48.573	15.128	18	75
Age_squared	Squared term of age	3654	2588.095	1439.85	324	5625
Working Characteristics						
ln_Income	Logarithm of personal total income (RMB)	3837	8.351	3.858	0	16.111
Whether working in the system	Yes = 1, No = 0	3851	0.065	0.247	0	1
Whether having pension	Yes = 1, No = 0	3853	0.718	0.450	0	1
Whether having medical insurance	Yes = 1, No = 0	3854	0.930	0.256	0	1
Human Capital Characteristics						
Education level	1–13 levels	3857	0.111	0.315	0	1
Health status	1–5 levels	3857	0.558	0.497	0	1
Social Characteristics						
Whether ethnic minorities	Yes = 1, No = 0	3857	0.075	0.264	0	1
Whether religious believer	Yes = 1, No = 0	3857	0.099	0.298	0	1
Whether CPC member	Yes = 1, No = 0	3855	0.101	0.301	0	1
Family Characteristics						
Family size	Number of members in the family	3855	2.921	1.595	1	30
Number of children	Number of children in the family	3852	1.592	1.243	0	22
Province dummies						

Notes: The education level is classified from 1 to 13: 1—without any education, 2—kindergarten, 3—primary school, 4—junior high school, 5—vocational high school, 6—ordinary high school, 7—technical secondary school, 8—technical high school, 9—junior college (adult education), 10—junior college (regular education), 11—undergraduate (adult education), 12—undergraduate (regular education), 13—postgraduate and above. Health status is based on the self-rated health levels from 1 to 5: 1—very unhealthy, 2—relatively unhealthy, 3—medium, 4—relatively healthy, 5—very healthy.

**Table 2 behavsci-12-00425-t002:** Benchmark results.

Model	(1) OLS	(2) OLS	(3) OLS	(4) OLS	(5) OLS	(6) OLS
Variable	Family Communication	Family Communication	Family Communication	Friends Communication	Friends Communication	Friends Communication
Internet usage	0.094 *** (0.026)	0.086 *** (0.027)	0.095 *** (0.028)	0.050 *** (0.012)	0.043 *** (0.011)	0.051 *** (0.012)
Whether the participant is female	1.544 ** (0.765)	1.133 (0.802)	1.396 * (0.807)	−0.306 (0.319)	−0.511 (0.330)	−0.506 (0.328)
Age	0.620 *** (0.164)	0.682 *** (0.176)	0.543 *** (0.180)	−0.177 ** (0.069)	−0.164 ** (0.073)	−0.147 * (0.076)
Age_squared	−0.005 *** (0.002)	−0.006 *** (0.002)	−0.004 ** (0.002)	0.002 ** (0.001)	0.002 ** (0.001)	0.002 ** (0.001)
ln_Income		−0.173 (0.114)	−0.092 (0.114)		0.013 (0.042)	0.040 (0.044)
Whether the participant is working in the system		−1.481 (1.360)	−1.250 (1.469)		−0.074 (0.597)	−0.125 (0.640)
Whether the participant has a pension		0.914 (0.956)	0.270 (0.988)		−0.828 ** (0.392)	−0.694 * (0.408)
Whether the participant has medical insurance		3.617 ** (1.486)	3.474 ** (1.491)		2.372 *** (0.529)	2.259 *** (0.531)
Education level			−1.816 (1.367)			0.246 (0.671)
Whether the participant is healthy			2.827 *** (0.852)			1.077 *** (0.355)
Whether the participant belongs to ethnic minorities			0.110 (1.703)			−0.754 (0.799)
Whether the participant is a religious believer			0.378 (1.397)			0.774 (0.650)
Whether the participant is a CPC member			0.468 (1.345)			−0.156 (0.612)
Family size			1.865 *** (0.382)			0.046 (0.095)
Number of children			−0.192 (0.353)			−0.225 (0.144)
Province dummies	No	No	Yes	No	No	Yes
Constant	3.642 (3.713)	0.648 (4.093)	−7.540 (4.852)	11.160 *** (1.646)	9.307 *** (1.801)	5.764 *** (2.042)
Observations	3740	3532	3527	3722	3514	3507

Notes: ***, **, and * indicate significance at the levels of 1%, 5%, and 10%, respectively. The values in parentheses are standard errors robust to heteroskedasticity. ‘Yes’ means the corresponding variables are controlled in the regression, while ‘No’ means they are not controlled.

**Table 3 behavsci-12-00425-t003:** Robustness checks using the Poisson model.

Model	(1) Poisson	(2) Poisson	(3) Poisson	(4) Poisson	(5) Poisson	(6) Poisson
Variable	Family Communication	Family Communication	Family Communication	Friends Communication	Friends Communication	Friends Communication
Internet usage	0.004 *** (0.001)	0.004 *** (0.001)	0.004 *** (0.001)	0.006 *** (0.001)	0.005 *** (0.001)	0.006 *** (0.001)
Demographic Characteristics	Yes	Yes	Yes	Yes	Yes	Yes
Working Characteristics	No	Yes	Yes	No	Yes	Yes
Human Capital Characteristics	No	No	Yes	No	No	Yes
Social Characteristics	No	No	Yes	No	No	Yes
Family Characteristics	No	No	Yes	No	No	Yes
Province dummies	No	No	Yes	No	No	Yes
Constant	2.226 *** (0.185)	2.076 *** (0.205)	1.753 *** (0.238)	2.481 *** (0.194)	2.194 *** (0.224)	1.702 *** (0.262)
Observations	3740	3532	3527	3722	3514	3507

Notes: *** indicate significance at the levels of 1%, respectively. The values in parentheses are standard errors robust to heteroskedasticity. ‘Yes’ means the corresponding variables are controlled in the regression, while ‘No’ means they are not controlled.

**Table 4 behavsci-12-00425-t004:** Robustness checks, using other indicators of interpersonal communications.

Model	(1) OLS	(2) Oprobit	(3) Ologit	(4) OLS	(5) Oprobit	(6) Ologit
Variable	Family Communication Frequency	Family Communication Frequency	Family Communication Frequency	Friends Communication Frequency	Friends Communication Frequency	Friends Communication Frequency
Internet usage	0.004 * (0.002)	0.002 ** (0.001)	0.004 * (0.002)	0.014 *** (0.002)	0.008 *** (0.001)	0.013 *** (0.002)
Controls	Yes	Yes	Yes	Yes	Yes	Yes
Province dummies	Yes	Yes	Yes	Yes	Yes	Yes
Constant	3.659 *** (0.461)			6.828 *** (0.380)		
Observations	3205	3205	3205	3217	3217	3217

Notes: ***, **, and * indicate significance at the levels of 1%, 5%, and 10%, respectively. The values in parentheses are standard errors robust to heteroskedasticity. ‘Yes’ means the corresponding variables are controlled in the regression, while ‘No’ means they are not controlled.

**Table 5 behavsci-12-00425-t005:** Endogeneity tests: impacts on communications using an instrumental variable.

Model	(1) First Stage	(2) 2SLS Second Stage	(3)First Stage	(4) 2SLS Second Stage
Variable	Internet Usage	Family Communication	Internet Usage	Friends Communication
Internet usage		0.606 ** (0.293)		0.249 ** (0.117)
Artificial Intelligence	7.440 *** (1.729)		7.704 *** (1.723)	
Controls	Yes	Yes	Yes	Yes
Province dummies	Yes	Yes	Yes	Yes
Constant	58.925 *** (6.060)	−44.955 ** (19.463)	56.252 *** (5.730)	−5.896 (7.888)
Observations	1889	1889	1880	1880

Notes: *** and ** indicate significance at the levels of 1% and 5%, respectively. The values in parentheses are standard errors robust to heteroskedasticity. ‘Yes’ means the corresponding variables are controlled in the regression, while ‘No’ means they are not controlled.

**Table 6 behavsci-12-00425-t006:** Replacing the missing values with the mean of the remaining values (OLS model).

Model	(1) OLS	(2) OLS	(3) OLS	(4) OLS	(5) OLS	(6) OLS
Variable	Family Communication	Family Communication	Family Communication	Friends Communication	Friends Communication	Friends Communication
Internet usage	0.094 *** (0.026)	0.098 *** (0.027)	0.111 *** (0.027)	0.050 *** (0.012)	0.051 *** (0.012)	0.058 *** (0.012)
Demographic Characteristics	Yes	Yes	Yes	Yes	Yes	Yes
Working Characteristics	No	Yes	Yes	No	Yes	Yes
Human Capital Characteristics	No	No	Yes	No	No	Yes
Social Characteristics	No	No	Yes	No	No	Yes
Family Characteristics	No	No	Yes	No	No	Yes
Province dummies	No	No	Yes	No	No	Yes
Constant	3.642 (3.713)	1.317 (3.952)	−6.838 (4.692)	11.160 *** (1.646)	9.288 *** (1.712)	5.222 *** (1.967)
Observations	3740	3740	3740	3740	3740	3740
Adjusted R^2^	0.009	0.011	0.051	0.012	0.014	0.036

Notes: *** indicate significance at the levels of 1%, respectively. The values in parentheses are standard errors robust to heteroskedasticity. ‘Yes’ means the corresponding variables are controlled in the regression, while ‘No’ means they are not controlled.

**Table 7 behavsci-12-00425-t007:** Further impacts on loneliness.

Model	(1) Oprobit	(2) Poisson	(3) Oprobit	(4) Poisson	(5) Oprobit
Variable	Lonely_1	Friends Communication	Lonely_1	Family Communication	Lonely_1
Internet usage	−0.003 ** (0.001)	0.006 *** (0.001)	−0.003 ** (0.001)	0.004 *** (0.001)	−0.002 * (0.001)
Friends communication			0.000 (0.002)		
Family communication					−0.005 *** (0.001)
Controls	Yes	Yes	Yes	Yes	Yes
Province dummies	Yes	Yes	Yes	Yes	Yes
Constant		1.702 *** (0.262)		1.753 *** (0.238)	
Observations	3615	3507	3499	3527	3518

Notes: ***, **, and * indicate significance at the levels of 1%, 5%, and 10%, respectively. The values in parentheses are standard errors robust to heteroskedasticity. ‘Yes’ means the corresponding variables are controlled in the regression, while ‘No’ means they are not controlled.

**Table 8 behavsci-12-00425-t008:** Other effects of internet usage (overlong usage and anxiety).

Model	(1) OLS	(2) OLS	(3) OLS	(4) OLS	(5) OLS	(6) OLS
Variable	Overtime Online	Anxiety When Offline	Less Outdoor Activities	Family Complaints	Vision Impairment	Neck and Shoulder Pain
Internet usage	0.010 *** (0.001)	0.008 *** (0.001)	0.007 *** (0.001)	0.006 *** (0.001)	0.008 *** (0.002)	0.008 *** (0.002)
Controls	Yes	Yes	Yes	Yes	Yes	Yes
Province dummies	Yes	Yes	Yes	Yes	Yes	Yes
Constant	3.502 *** (0.267)	2.940 *** (0.275)	3.164 *** (0.287)	3.847 *** (0.280)	3.373 *** (0.305)	2.942 *** (0.310)
Observations	2198	2206	2206	2204	2200	2203
Ajusted/Pseudo R^2^	0.105	0.080	0.101	0.103	0.055	0.038

Notes: *** indicate significance at the levels of 1%, respectively. The values in parentheses are standard errors robust to heteroskedasticity. ‘Yes’ means the corresponding variables are controlled in the regression, while ‘No’ means they are not controlled.

**Table 9 behavsci-12-00425-t009:** Heterogeneity analysis, in terms of online contacts.

Model	(1) OLS	(2) OLS	(3) OLS	(4) OLS
Sample	Less Online Social Contact	More Online Social Contact	Less Online Social Contact	More Online Social Contact
Variable	Family Communication	Family Communication	Friends Communication	Friends Communication
Internet usage	0.055 (0.081)	0.105 *** (0.031)	0.033 (0.025)	0.053 *** (0.013)
Controls	Yes	Yes	Yes	Yes
Constant	10.825 (12.224)	−6.455 (5.955)	−0.647 (4.453)	10.655 *** (2.724)
Observations	1701	1826	1685	1822

Notes: *** indicate significance at the levels of 1%, respectively. The values in parentheses are standard errors robust to heteroskedasticity. ‘Yes’ means the corresponding variables are controlled in the regression, while ‘No’ means they are not controlled.

**Table 10 behavsci-12-00425-t010:** Heterogeneity analysis in terms of online posts.

Model	(1) OLS	(2) OLS	(3) OLS	(4) OLS
Sample	Fewer Online Posts	More Online Posts	Fewer Online Posts	More Online Posts
Variable	Family Communication	Family Communication	Friends Communication	Friends Communication
Internet usage	0.050 (0.047)	0.129 *** (0.037)	0.036 ** (0.019)	0.051 *** (0.015)
Controls	Yes	Yes	Yes	Yes
Constant	0.503 (7.625)	−12.146 * (7.119)	4.028 (2.927)	7.587 ** (3.146)
Observations	2284	1242	2260	1246

Notes: ***, **, and * indicate significance at the levels of 1%, 5%, and 10%, respectively. The values in parentheses are standard errors robust to heteroskedasticity. ‘Yes’ means the corresponding variables are controlled in the regression, while ‘No’ means they are not controlled.

**Table 11 behavsci-12-00425-t011:** Heterogeneity analysis, in terms of Internet skills.

Model	(1) OLS	(2) OLS	(3) OLS	(4) OLS
Sample	Less Skilled in Internet	More Skilled in Internet	Less Skilled in Internet	More Skilled in Internet
Variable	Family Communication	Family Communication	Friends Communication	Friends Communication
Internet usage	0.025 (0.061)	0.118 *** (0.033)	0.041 * (0.024)	0.049 *** (0.014)
Controls	Yes	Yes	Yes	Yes
Constant	24.445 (16.486)	−10.005 (6.167)	−6.478 (4.713)	10.110 *** (2.915)
Observations	1836	1675	1821	1670

Notes: *** and * indicate significance at the levels of 1% and 10%, respectively. The values in parentheses are standard errors robust to heteroskedasticity. ‘Yes’ means the corresponding variables are controlled in the regression, while ‘No’ means they are not controlled.

**Table 12 behavsci-12-00425-t012:** Heterogeneity analysis, in terms of age.

Model	(1) OLS	(2) OLS	(3) OLS	(4) OLS
Sample	Younger than 35	Older than 35	Younger than 35	Older than 35
Variable	Family Communication	Family Communication	Friends Communication	Friends Communication
Internet usage	0.133 *** (0.045)	0.058 * (0.035)	0.059 *** (0.020)	0.039 *** (0.013)
Controls	Yes	Yes	Yes	Yes
Constant	2.054 (4.982)	12.371 *** (3.118)	6.489 *** (1.970)	1.786 * (1.079)
Observations	842	2685	845	2662

Notes: *** and * indicate significance at the levels of 1% and 10%, respectively. The values in parentheses are standard errors robust to heteroskedasticity. ‘Yes’ means the corresponding variables are controlled in the regression, while ‘No’ means they are not controlled.

**Table 13 behavsci-12-00425-t013:** Heterogeneity analysis, in terms of education level.

Model	(1) OLS	(2) OLS	(3) OLS	(4) OLS
Sample	Lower Education	Higher Education	Lower Education	Higher Education
Variable	Family Communication	Family Communication	Friends Communication	Friends Communication
Internet usage	0.077 ** (0.031)	0.147 ** (0.060)	0.044 *** (0.012)	0.089 *** (0.034)
Controls	Yes	Yes	Yes	Yes
Constant	−4.015 (5.388)	−9.295 (14.546)	3.895 * (2.298)	12.099 ** (5.454)
Observations	3141	386	3123	384

Notes: ***, **, and * indicate significance at the levels of 1%, 5%, and 10%, respectively. The values in parentheses are standard errors robust to heteroskedasticity. ‘Yes’ means the corresponding variables are controlled in the regression, while ‘No’ means they are not controlled.

**Table 14 behavsci-12-00425-t014:** Heterogeneity analysis in terms of region.

Model	(1) OLS	(2) OLS	(3) OLS	(4) OLS
Sample	Rural Residents	Urban Residents	Rural Residents	Urban Residents
Variable	Family Communication	Family Communication	Friends Communication	Friends Communication
Internet usage	0.086 ** (0.041)	0.096 ** (0.038)	0.041 *** (0.014)	0.058 *** (0.018)
Controls	Yes	Yes	Yes	Yes
Constant	1.852 (6.737)	−12.355 (7.981)	4.046 * (2.349)	9.693 ** (3.826)
Observations	2235	1280	2226	1269

Notes: ***, **, and * indicate significance at the levels of 1%, 5%, and 10%, respectively. The values in parentheses are standard errors robust to heteroskedasticity. ‘Yes’ means the corresponding variables are controlled in the regression, while ‘No’ means they are not controlled.

**Table 15 behavsci-12-00425-t015:** Heterogeneity analysis in terms of migration.

Model	(1) OLS	(2) OLS	(3) OLS	(4) OLS
Sample	Non-Migrants	Migrants	Non-Migrants	Migrants
Variable	Family Communication	Family Communication	Friends Communication	Friends Communication
Internet usage	0.077 ** (0.035)	0.112 ** (0.045)	0.055 *** (0.016)	0.037 ** (0.017)
Controls	Yes	Yes	Yes	Yes
Constant	−5.386 (6.598)	−10.688 (7.774)	1.386 (2.558)	10.472 *** (3.601)
Observations	2422	1094	2406	1089

Notes: *** and ** indicate significance at the levels of 1% and 5%, respectively. The values in parentheses are standard errors robust to heteroskedasticity. ‘Yes’ means the corresponding variables are controlled in the regression, while ‘No’ means they are not controlled.

## Data Availability

The data that support the findings of this study are available from the Chinese General Social Survey (CGSS, http://cgss.ruc.edu.cn/English/Home.htm (accessed on 25 September 2022). Restrictions apply to the availability of these data, which were used under license for this study. Data are also available from the authors with the permission of the CGSS.

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
