# Peer review of "Does the Internet Bring People Closer Together or Further Apart? The Impact of Internet Usage on Interpersonal Communications"

_behavsci, 2022, doi:10.3390/bs12110425_

Round 1

Reviewer 1 Report

The Authors correctly defined the purpose of the paper, adequately identified gaps in the area of research and constructed research hypotheses.

The review of the literature, especially research in this area, is up-to-date and includes recent analytics on the impact of the Covid-19 pandemic on interpersonal communication. The authors considered both the favourable and unfavourable influence of the Internet on interpersonal communication.

The analysis was carried out clearly and logically. Even though I do not find errors in the performed statistical analysis or the presented conclusions, I have an impression of a lack of some further implications. Above all, I miss references or attempts to indicate how Internet usage changes face-to-face interpersonal communication. The Authors show a positive impact of Internet communication on interpersonal relations - but it isn't easy to assume that it has only a positive dimension. Assuming that the aim of the analysis has been achieved, an important question arises, which the Authors did not ask, first of all, does communication on the Internet affect everyday face-to-face conversations?; does it affect the quality of offline interpersonal relationships? (for example, change of conversation topics, reluctance to take up topics previously discussed on the Internet, unwillingness to reveal one's true thoughts in a face-to-face relationship etc.). The authors rightly state in verses 42-44 that 'the impact of Internet usage on interpersonal communications is still unclear.' In the article's summary, it is worth indicating other areas that require clarification and analysis in the future (7. Conclusion and Discussion).

Reviewer 2 Report

The most visible weakness of this paper is its broad topical application/abstract context. Studying the effect of Internet usage on interpersonal connections has merit but the Internet itself, in its very inherent characteristic, has diverse and varying uses, each for a different purpose. Apps existing solely for communication or establishing social connections (i.e., social platforms) may contribute to a positive correlation to interpersonal connections, whereas software not purposed for sustaining relationships, such as streaming apps and productivity tools, may negatively affect people’s interpersonal connections. Which of the two streams is the focus of this paper? The authors never distinguished what the context of the term Internet usage may refer to. This is the case in the whole paper, including the dataset used, CGSS, and internet-usage habits (the exploratory IV) of the respondents who never elaborated on what sites/apps they are really using to establish a link on the DV (interpersonal connection) of this paper.

Reviewer 3 Report

This is an interesting study examining the impact of internet usage on interpersonal interactions. The paper is well-written. I believe that it will contribute to the literature well. I have several comments to improve the manuscript further:

1. It will be useful for the authors to provide more information on the Chinese General Social Survey. What is the sampling method of the study? What is the procedure? As it is mentioned that the sample is a nationally representative sample, there is a need for more justification in this issue 

2. The missing data and data imputation technique should be highlighted in the method section  

Ibrahim, J. G., & Molenberghs, G. (2009). Missing data methods in longitudinal studies: A review. Test, 18(1), 1-43. 

3. The discussion section seems to be too short. There is a lack of discussion on potential implication of the findings. The literature review was done well in the Introduction but the quality of the discussion need to be improved further. There is a need to relate the current findings to the existing studies at least.

4. One important aspect to discuss is the use of subjective measures of time spent on using the internet as well as daily interaction. Recent studies have suggested that subjective measure tend to be bias and inaccurate. It will be important for the authors to discuss and acknowledge this limitation

Relevant paper: Hodes, L. N., & Thomas, K. G. (2021). Smartphone screen time: inaccuracy of self-reports and influence of psychological and contextual factors. Computers in Human Behavior, 115, 106616.   5. In the discussion, I would suggest the authors to discuss its finding in relation to existing studies that found the positive relation between internet use and well-being. The finding of the current study is important and can contribute to the literature by elucidating the mechanisms underlying the well-being and emotional benefits of internet use, especially in older adults (see the following paper):   Relevant paper to discuss:Cognitive, social, emotional, and subjective health benefits of computer use in adults: A 9-year longitudinal study from the Midlife in the United States (MIDUS). (2020). Computers in Human Behavior, 104, 106179.     6. Lastly, it will be important for the authors to highlight some of the limitations of the study in the discussion.

Reviewer 4 Report

The main aim of the paper was to examine the influence of Internet usage on interpersonal communications using data from the Chinese General Social Survey to consider whether the Internet brings people closer together or further apart. The findings show that Internet usage assists to increase the time and communications frequency with family and friends. Internet usage was also found to contribute to decreased loneliness, and improved interactions with family members. The positive role of Internet usage on communications was also found to be more prominent for people with more frequent online socialization and self-presentation, higher online skills, younger age, higher educational levels and living in urban areas. The beneficial impact of Internet usage was also revealed to be larger on communications with family members for migrants. These findings make an original contribution to Internet usage on interpersonal communications discourse.

However, the theoretical foundation on which the research was based was lacking and should be introduced in the Introduction and then significantly expanded in the literature review. The authors should take as a starting-point one or more sufficiently contrasted theories and apply them to this new context of analysis to justify the need to develop this new research. The paper should incorporate a more solid argumentation that allows justification of the reason for the selection of the explanatory variables that are considered in the empirical analysis. Stating that there is a lack of theoretical explanations and empirical evidence is an insufficient reason/justification to conduct the research. Furthermore, as far as possible, the theoretical framework should be sufficiently solid to justify that the relevant variables that should explain the phenomenon under study are those considered in the analysis and only those variables. In short, why is this research necessary; why are you using the theoretical model; what are the research gaps; and what recent justification have you provided for the aforementioned? In summary, fully explain the theoretical framework (in the Introduction and Literature review sections) that served as the foundation for your conceptual model that you developed.

Additionally, the Introduction section should not include the hypotheses, which should be included in a separate section after the Theoretical Framework/Underpinning, which should be positioned after the Literature Review section.

I did not see a section dedicated to the theoretical framework/underpinning on which your research was based. The theoretical framework is used to explain proven theories that embody the findings of a number of investigations of phenomena under different scenarios. The theoretical framework frames the study and identifies the key theories and concepts that underpin your study, so this needs to be outlined in relation to your research objectives. You also listed two of hypotheses, but these should also include suitable substantiation or justification. Hence, each individual hypothesis should be substantiated/justified by suitable literature, and positioned after the Theoretical Framework/Underpinning section!

The paper explores relevant literature, and there were a number of suitable sources from 2020 – 2022, which were incorporated throughout the paper to support the research.

The Methods (Data and Measures) section was apt.

The Results and Discussion sections were appropriately presented.

The Conclusion and Recommendations section was much too brief, should also include separate Theoretical and Managerial or Practical Implications’ sections:

Theoretical implications: What contribution has the research has made to theory based on the theoretical foundation/model that you selected (or in your case did not select).

Managerial or practical implications: Explain the contribution that the research has made in terms of managerial or practical implications, i.e. how can companies, managers and organizations use your research?

You also need to provide a separate limitations (and directions for future research) section, which details possible limitations (hindrances) that may influence the research process, and in term of  methodological, coverage and generalizability limitations, as well as risks to credibility of your research. Additional research could be proposed to over these limitations.

There are some grammatical/language issues in the paper, so the paper is in need of a language expert or grammarian to language edit the paper. For example, page 1 line 26 “well-ness” should be “wellness”.

Overall, an interesting paper, which makes an original contribution, but I suggest some major revision before the paper can be accepted for publication.

Round 2

Reviewer 2 Report

I accept the comments and revisions.

Author Response

Dear Reviewer: Thank you so much again for your valuable comments which allowed us to greatly improve the quality of this manuscript. Many thanks! Sincere regards, All the authors

Reviewer 3 Report

The authors have addressed my comments very well. Very impressive. I appreciate all their efforts.

Author Response

(The authors gave the same response as above.)

Reviewer 4 Report

The main aim of the paper was to examine the influence of Internet usage on interpersonal communications using data from the Chinese General Social Survey to consider whether the Internet brings people closer together or further apart via the Complementarity-Interference model. The findings show that Internet usage assists to increase the time and communications frequency with family and friends. Internet usage was also found to contribute to decreased loneliness, and improved interactions with family members. The positive role of Internet usage on communications was also found to be more prominent for people with more frequent online socialization and self-presentation, higher online skills, younger age, higher educational levels and living in urban areas. The beneficial impact of Internet usage was also revealed to be larger on communications with family members for migrants. These findings make an original contribution to Internet usage on interpersonal communications discourse.

My comment relating to the lack of a theoretical foundation on which the research was based and that the authors should provide sufficiently contrasted theories and apply them to this new context of analysis to justify the need to develop this new research, as well as incorporate a more solid argumentation that allows justification of the reason for the selection of the explanatory variables that are considered in the empirical analysis has been suitably addressed. The authors supplemented the introduction and literature review with the analysis of the theoretical foundation of this study, namely via the CI model of Internet was applied as the theoretical foundation. A new Theoretical Framework section was included, where the theories were to exemplify the findings of a number of investigations of phenomena under different scenarios. The section identified the key theories and concepts that underpin the research. The authors also explained the reasons for the selection of the core explained and explanatory variables based on the existing literature and theoretical analysis. So my comment concerning the lack of theoretical explanations and empirical evidence is an insufficient reason/justification to conduct the research, and that the theoretical framework should be sufficiently solid to justify that the relevant variables that should explain the phenomenon under study are those considered in the analysis and only those variables was aptly addressed.

My comment “In short, why is this research necessary; why are you using the theoretical model; what are the research gaps; and what recent justification have you provided for the aforementioned? In summary, fully explain the theoretical framework (in the Introduction and Literature review sections) that served as the foundation for your conceptual model that you developed” was also adequately addressed. The complementary and interference aspects of the Internet affecting interpersonal communications were explored by the authors, from which the gap of existing research, the research theme of this paper, and its significance were clarified.

My comment relating to the Introduction section not including the hypotheses, and should be included in a separate section after the Theoretical Framework/Underpinning was also aptly addressed. The authors removed the hypotheses from the introduction and put it in the theoretical analysis section, where each individual hypothesis was substantiated by existing literature.

The paper explores relevant literature, and there were a number of suitable sources from 2020 – 2022, which were incorporated throughout the paper to support the research.

The Methods (Data and Measures) section was apt.

The Results and Discussion sections were appropriately presented.

My comments relating to The Conclusion and Recommendations section being much too brief, and should also include separate Theoretical and Managerial or Practical Implications’ sections were also suitably remedied. The authors rewrote a separate conclusion section and summarized the findings in more detail. They also added a new section to discuss theoretical and practical implications.

My comment about proving a separate limitations (and directions for future research) section was also adequately addressed by the authors. The authors discussed the limitations of this paper and provided valuable directions for further research based on the shortcomings.

The grammatical/language issues in the paper seem to be improved, but additional language editing prior to publication would be beneficial.  

Overall, an interesting paper, which makes an original contribution, and I recommend that it can be accepted for publication after some more language editing to make it easier to read and enhance the transfer of knowledge to the reader.

Author Response

Dear Reviewer:

Thank you so much for your valuable additional comments concerning the manuscript entitled “Does Internet Bring People Closer Together or Further Apart?: Impact of Internet Usage on Interpersonal Communications”. Your comments that “the grammatical/language issues in the paper seem to be improved, but additional language editing prior to publication would be beneficial” are very helpful to remind us to further proofread and polish this manuscript. Thorough, thoughtful and strategic editing has been conducted to improve the flow and readability of the manuscript, with the help of an experienced English editor.

Furthermore, following the academic editor’s suggestions, the contents throughout the manuscript are simplified to highlight the key points and the research logic, especially the relations among the research questions in this study, is more clearly pointed out.

Further revisions made to the manuscript are marked up using the “Track Changes” function in MS Word, such that changes can be easily viewed. We hope that all these revisions could meet with your approval for publication. Again, in any case, we are open to any further comments on this manuscript.

Many thanks!

Sincere regards,

All the authors